# Data-Free One-Shot Federated Regression:
# An Application to Bone Age Assessment

**Zhou Zheng**[1,*]                                    ZZHENG@MORI.M.IS.NAGOYA-U.AC.JP
**Yuichiro Hayashi**[1]                                YHAYASHI@MORI.M.IS.NAGOYA-U.AC.JP
**Masahiro Oda**[1]                                    MODA@MORI.M.IS.NAGOYA-U.AC.JP
[1] *Nagoya University, Japan*

**Takayuki Kitasaka**[2]                               KITASAKA@AITECH.AC.JP
[2] *Aichi Institute of Technology, Japan*

**Kensaku Mori**[1,3,*]                                KENSAKU@IS.NAGOYA-U.AC.JP
[1] *Nagoya University, Japan* [3] *National Institute of Informatics, Japan*

## Abstract

We consider a novel problem setting: data-free one-shot federated regression. This setting aims to prepare a global model through a single round of communication without relying on auxiliary information, *e.g.*, proxy datasets. To address this problem, we propose a practical framework that consists of three stages: local training, data synthesizing, and knowledge distillation, and demonstrate its efficacy with an application to bone age assessment. We conduct validation under independent and identical distribution (IID) and non-IID settings while considering both model homogeneity and heterogeneity. Validation results show that our method surpasses `FedAvgOneShot` by a large margin and sometimes even outperforms the proxy-data-dependent approach `FedOneShot`.

**Keywords:** Federated learning, Regression, One-shot, Data-free.

## 1. Introduction

One-shot federated learning (FL) (Guha et al., 2019) has emerged as a potential solution to address concerns regarding the costly inter-node communication and possible privacy leakage in standard FL methods, as it allows for only a single global round between clients and the central server. While one-shot FL methods typically require additional sources like proxy datasets for global model training, recent advances in data-free one-shot FL (Zhang et al., 2022; Luz-Ricca et al., 2023) overcomes this limitation, eliminating the need for additional datasets. Nevertheless, current FL methods have predominantly focused on classification tasks, with a limited exploration of regression problems.

Motivated by these observations, we consider a novel problem setting: data-free one-shot federated regression. Inspired by the work (Zhang et al., 2022) that proposed for classification, we present a practical framework specialized for regression, which comprises three stages: **local training**, **data synthesizing**, and **knowledge distillation** (KD), and evaluate it with a bone age assessment task (Halabi et al., 2019). Our method is the first attempt in this setting, and validation results demonstrate its efficacy.

---

\* Send correspondence to Zhou Zheng or Kensaku Mori. This work was supported by JSPS KAKENHI Grant Numbers 21K19898 and 17H00867 and JST CREST Grant Number JPMJCR20D5, Japan.

## 2. Method

Generally, let there be $K$ local clients $\{C_i\}_{i=1}^K$ with each client holding a private dataset $\mathbf{D}_i = \{\mathbf{x}_i, y_i\}$, where $\mathbf{x}_i$ are images, and $y_i$ are ground truth, *e.g.*, bone ages in our study.

**First stage: local training.** Each client $C_i$ trains its local model $M_i(\cdot, \boldsymbol{\Theta}i)$ with the private dataset $\mathbf{D}_i$ and uploads the model weight to the central server after training.

**Second stage: data synthesizing.** We adopt a generator $G(\cdot, \boldsymbol{\Theta}g)$ to synthesize images. In our study, bone ages range from 1 to 228 months, and we assume $y$ follows a discrete uniform distribution $p(y)$ over the set of $\{1, 2, 3, ..., 228\}$. To train $G(\cdot, \boldsymbol{\Theta}g)$, we first sample a batch of random noise vectors $\mathbf{z} \sim N(\mathbf{0}, \mathbf{I})$ and a batch of random values $\hat{y} \sim p(y)$. Afterward, we input $\mathbf{z}$ to $G(\cdot, \boldsymbol{\Theta}g)$ to get a batch of produced images $\hat{\mathbf{x}} = G(\mathbf{z}, \boldsymbol{\Theta}g)$. Next, we feed $\hat{\mathbf{x}}$ into local models $\{M_i(\cdot, \boldsymbol{\Theta}i)\}_{i=1}^K$ to get predictions $\{M_i(\hat{\mathbf{x}}, \boldsymbol{\Theta}i)\}_{i=1}^K$. By adopting the basic ensemble scheme (Mendes-Moreira et al., 2012), we get the ensembled result of local models $E(\hat{\mathbf{x}}) = \sum_1^K M_i(\hat{\mathbf{x}}, \boldsymbol{\Theta}i)/K$. We calculate a loss $\mathcal{L}_{sim}(E(\hat{\mathbf{x}}), \hat{y}) = \|E(\hat{\mathbf{x}}) - \hat{y}\|_2$ to expect $\hat{\mathbf{x}}$ following a similar distribution to $\mathbf{x}$. Besides, to improve the quality of $\hat{\mathbf{x}}$, we adopt a feature distribution regularization term $\mathcal{L}_{feat}$ (Yin et al., 2020), which enforces feature-level similarity and is defined as $\mathcal{L}_{feat}(\hat{\mathbf{x}}) = \sum_{k=1}^K \sum_l \left( \|\mu_{k,l}(\hat{\mathbf{x}}) - \mu_{k,l}(\mathbf{x})\|_2 + \|\sigma_{k,l}(\hat{\mathbf{x}}) - \sigma_{k,l}(\mathbf{x})\|_2 \right)/K$, where $\mu_{k,l}(\cdot)$ and $\sigma_{k,l}(\cdot)$ denote the mean and variance of features of $l$-th batch normalization layer for $M_i(\cdot, \boldsymbol{\Theta}i)$. In addition, to ensure $G(\cdot, \boldsymbol{\Theta}g)$ generates more diverse images, we propose $\mathcal{L}_{dis}$ to encourage disagreement between local models $\{M_i(\cdot, \boldsymbol{\Theta}i)\}_{i=1}^K$ and the global model $S(\cdot, \boldsymbol{\Theta}s)$, which is written as $\mathcal{L}_{dis}(\hat{\mathbf{x}}) = -\|E(\hat{\mathbf{x}}) - S(\hat{\mathbf{x}}, \boldsymbol{\Theta}s)\|_2$. To conclude, the total training objective of $G(\cdot, \boldsymbol{\Theta}g)$ is $\mathcal{L}_{gen}(\hat{\mathbf{x}}, \hat{y}) = \mathcal{L}_{sim}(E(\hat{\mathbf{x}}), \hat{y}) + \lambda \mathcal{L}_{feat}(\hat{\mathbf{x}}) + \beta \mathcal{L}_{dis}(\hat{\mathbf{x}})$. We set $\lambda$ to 0.5 and $\beta$ to 0.1. Note that local and global models are fixed at this stage.

**Third stage: knowledge distillation.** We update the global model $S(\cdot, \boldsymbol{\Theta}s)$ by knowledge transfer. Specifically, the fixed generator first synthesizes a batch of images $\hat{\mathbf{x}}$ when feeding a batch of random noise vectors $\mathbf{z}$. Then $\hat{\mathbf{x}}$ are input into local models to get ensembled prediction $E(\hat{\mathbf{x}})$. We finally utilize a loss $\mathcal{L}_{kd}(E(\hat{\mathbf{x}}), S(\hat{\mathbf{x}}, \boldsymbol{\Theta}s)) = \|E(\hat{\mathbf{x}}) - S(\hat{\mathbf{x}}, \boldsymbol{\Theta}s)\|_2$ to enforce the similarity between $E(\hat{\mathbf{x}})$ and $S(\hat{\mathbf{x}}, \boldsymbol{\Theta}s)$.

## 3. Experiments, Results, and Conclusions

**Dataset and metric.** We applied the public dataset RNSA-BAA (Halabi et al., 2019), which contains 12,611/1,425/200 hand radiographs for training/validation/testing. We reported the mean absolute difference (MAD) results on the test set based on three runs.

**Experimental setup.** We maintained four local clients. We divided the training set into four subsets with bone age values falling within four ranges, as shown in Figure 1(a). To simulate an IID setting among clients, we ensured that each client received a similar number of images within the same bone age range by randomly extracting 1/4 of the data from each subset without repetition and assigning them to individual clients, as shown in Figure 1(b). Conversely, to form a non-IID setting, we distributed one subset to one client, as illustrated in Figure 1(c). We also considered model homogeneity and heterogeneity. Thus, we introduced four different settings: (1) **homo-IID**: model homogeneity with IID. (2) **homo-non-IID**: model homogeneity with non-IID. (3) **hetero-IID**: model heterogeneity with IID. (4) **hetero-non-IID**: model heterogeneity with non-IID.

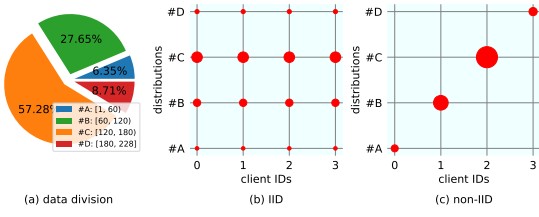

Figure 1: Details of experimental setup: (a) Training set was divided into four subsets with bone age values falling within four ranges. (b) Simulated independent and identically distributed (IID) setting. (c) Simulated non-IID setting. Size of each red circle is proportional to number of samples.

Table 1: Quantitative comparison among different methods under four different settings. `Centralization` represents the upper-bound accuracy derived by centralized training. Results are reported as average (standard deviation) on test set based on three runs. '↓': lower values of mean absolute difference (MAD) indicate better performance. '-': results are not applicable.

| method | MAD ↓ | | | |
|---|---|---|---|---|
| centralization | 10.15 (0.46) | | | |
| | homo-IID | homo-non-IID | hetero-IID | hetero-non-IID |
| FedAvg | 11.68 (0.48) | 36.80 (1.23) | - | - |
| FedAvgOneShot | 62.15 (2.75) | 68.35 (2.75) | - | - |
| FedNoisyKD | 116.41 (1.78) | 116.46 (1.57) | 117.15 (2.07) | 117.59 (2.44) |
| FedOneShot | 59.92 (3.31) | **46.55 (1.69)** | **55.87 (2.67)** | **46.29 (1.10)** |
| Ours | **42.65 (4.40)** | 49.49 (5.45) | 58.52 (30.01) | 52.60 (5.98) |

**Baselines.** We compared our scheme with `FedAvg` (McMahan et al., 2017) and its one-shot version `FedAvgOneShot` (averaging model weights after local training). We also implemented a scheme that used random noise images for KD, and we abbreviated it as `FLNoisyKD`. In addition, we realized `FedOneShot` (Guha et al., 2019) using a public dataset (Pietka et al., 2001) as the proxy dataset for KD.

**Implementation details.** When considering model homogeneity, all clients used ResNet34 (He et al., 2016). For model heterogeneity, we applied ResNet34, ResNet50, and two variants (WRN-16-10, WRN-40-6) based on Wide-Resnets (Zagoruyko and Komodakis, 2016). The global model was always adopted as ResNet34. We applied the Adam optimizer. Local models were trained for 200 epochs using a poly-learning rate with an initial value of $10^{-4}$. We set a learning rate of $10^{-3}$ and the latent dimension of $\mathbf{z}$ to train the generator to 128. We set an initial learning rate of $10^{-4}$ to train the global model and decayed it to $10^{-6}$. The generator and the global model were trained in a loop of 120 rounds, and at each round, we trained the generator for 40 epochs and the global model for 1 epoch. We set the batch size to 32. All images were resized to a size of $224 \times 224$ pixels.

**Experiment results.** As illustrated in Table 1, `Centralization` represents the upper-bound accuracy derived by centralized training. We can observe a noticeable accuracy gap between `FedAvg` with IID and non-IID, indicating that `FedAvg` is also sensitive to non-IID in regression, similar to classification (Hsu et al., 2019). Then let us focus on one-shot FL methods. Limiting by a single global round, `FedAvgOneShot` achieves much larger MAD values than `FedAvg`. `FedNoisyKD` uses random noise images for KD, leading to the worst performance. `FedOneShot`, which conducts KD with a public dataset, achieves overall the best results. Compared to `FedOneShot`, our method outperforms it under the setting of homo-IID and realizes competitive accuracy under the other three settings. This suggests that our approach has the potential to synthesize images comparable to authentic images for KD, eliminating the requirements for proxy datasets.

**Conclusions.** This paper made a first attempt to explore data-free one-shot FL in regression. Our method demonstrated its efficacy in this setting. Future work may investigate improving image generation and apply the proposed method to more regression tasks.

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
