# OpenReview forum: "Data-Free One-Shot Federated Regression: An Application to Bone Age Assessment"
_MIDL.io/2023/Short_Paper_Track — MIDL 2023 Short paper track Poster_

### Official Review · Reviewer_AbQ4 · 2023-04-20
**Interesting contribution with rich experiments**

**Rating:** 8
**Confidence:** 3

**Review:**

Summary
Inspired by (Zhang et al., 2022), this work modified the original classification framework and proposed a data-free one-shot regression framework. Using a bone age regression task, the authors demonstrated the performance of the proposed method compared with baselines in different problems settings,

Strengths
(1) The paper is generally easy to read and follow, and well introduced and described the non-trivial investigation of data-free one-shot regression problems.
(2) Rich experiments demonstrated the good performance of the proposed methods compared with baselines. The proposed data-free method sometimes even outperforms baselines using proxy dataset.

Weaknesses
A diagram or a pseudo algorithm of the proposed methods would be better for readers to understand how the three stages work together in one-shot federated learning.

---

### Official Review · Reviewer_DUF5 · 2023-04-25
**Application of interesting data-free one-shot federated learning approach to bone age estimation**

**Rating:** 7
**Confidence:** 4

**Review:**

This paper proposes a data-free one-shot federated learning method for regression applications. Local models are first trained, then a data generator to eliminate need for a proxy dataset, and finally knowledge distillation is used with the generated images to transfer learning from ensemble of local models to the global model. Experiments are conducted for bone age estimation application under different distribution and model settings and compared against FedAvg and one-shot baselines.

Strengths:
+ Interesting area of research, important to combine datasets from multiple sources while addressing privacy leakage concerns
+ Reproducibility of paper is good given clarity of paper, included details, and use of public dataset

Weaknesses:
- While authors attribute "inspiration" of approach to Zhang et al, 2022 (which is cited), I do not see any methodological differences in the proposed method compared to Zhang et al, other than this work is applied to bone age regression problem. As an application paper I think it is good work, but the credit to prior work should be rightly attributed.
- The proposed method appears to have much larger variation in model output than compared approaches (Table 1, eg., hetero-IID std is 30 compared to other methods which is < 3) . Even though the MAD is overall reduced, this instability is concerning.

Minor points:
- typo for feature-level similarity loss L_feaSt --> L_feat
- setting (2) homo with non-IID when defined is missing _non_-IID